# Potent and Non-Cytotoxic Antibacterial Compounds against Methicillin-Resistant *Staphylococcus aureus* Isolated from *Psiloxylon mauritianum*, A Medicinal Plant from Reunion Island

**DOI:** 10.3390/molecules25163565

**Published:** 2020-08-05

**Authors:** Jonathan Sorres, Amandine André, Elsa Van Elslande, Didier Stien, Véronique Eparvier

**Affiliations:** 1Association DESIBER, 98 rue Roger Payet, Rivière des Pluies, La Réunion, 97438 Sainte Marie, France; 2CNRS, Institute of Chemistry of Natural Substances UPR2301, University of Paris-Saclay, 91198 Gif-sur-Yvette, France; andramandine@gmail.com (A.A.); elsa.van-elslande@cnrs.fr (E.V.E.); 3Laboratoire Shigeta, 62 boulevard Davout, 75020 Paris, France; 4Laboratory of Biodiversity and Microbial Biotechnologies (LBBM), Sorbonne University, CNRS, 75006 Paris, France; UPMC Univ Paris 06, Banyuls-sur-Mer Oceanological Observatory, 66650 Banyuls-sur-Mer, France; didier.stien@cnrs.fr

**Keywords:** Aspidin derivatives, acylphloroglucinol, *Psiloxylon mauritianum*, *Staphylococcus aureus*, antibacterial compounds

## Abstract

With the occurrence of antibiotic-resistant *Staphylococcus aureus* strains, identification of new anti-staphylococcal drugs has become a necessity. It has long been demonstrated that plants are a large and diverse source of antibacterial compounds. *Psiloxylon mauritianum*, an endemic medicinal plant from Reunion Island, was chemically investigated for its reported biological activity against *S. aureus*. Aspidin VB, a phloroglucinol derivative never before described, together with Aspidin BB, were first isolated from the ethyl acetate extract of *P. mauritianum* leaves. Their structures were elucidated from spectroscopic data. Aspidin VB exhibited strong antibacterial activity against standard and methicillin-resistant *S. aureus* strains, with a minimal inhibition concentration (MIC) of 0.25 μg/mL, and no cytotoxicity was observed at 10^−5^ M in MRC5 cells. Due to its biological activities, Aspidin VB appears to be a good natural lead in the fight against *S. aureus*.

## 1. Introduction

*Staphylococcus aureus* is a Gram-positive bacterium and the major cause of hospital-acquired infections, often resulting in longer stays and increases in patient mortality [1]. Such *S. aureus* infections, promoted by the use of ventilators or venous catheters, affect the bloodstream, lower respiratory tract, and the skin and soft tissues [2]. The microbial world is ruled by adaptation to environmental pressure, and *S. aureus* has developed very effective tools to resist antibiotics since the introduction of penicillin in the 1940s to cure infections. The selective pressure of antibiotics continually promotes the emergence of drug-resistant strains of *S. aureus*, which have dramatically increased and spread around the world [3].

Methicillin-resistant *Staphylococcus aureus* (MRSA) emerged quickly after introduction of the first semi-synthetic β-lactam in 1961 and has become a major worldwide health care problem [2]. Due to the rapidity and extent of its spread, as well as the high diversity among clones and strain virulence, the WHO has classified MRSA as a high priority target for new antibiotic development [4].

Even if pharmaceutical companies prefer combinatorial chemistry library strategies, the large diversity of natural products offers a wide range of antimicrobials [5]. Plant sources of anti-staphylococcal compounds should be highlighted due to reports in the literature of remarkable activities of acylphloroglucinols or terthiophenes, which have minimum inhibitory concentrations (MIC) of less than 1 μg/mL [5].

*Psiloxylon mauritianum* Baill. is a dioecious glabrous flowering plant classified as a member of the Myrtaceae family and is a unique species of the genus *Psiloxylon* [6]. *P. mauritianum* is endemic to Reunion Island and Mauritius and used there as a medicinal plant for the treatment of common infectious and inflammatory diseases, hypercholesterolemia, gout, dysentery and to alleviate symptoms of amenorrhea [7,8]. In 2013, the leaves of *P. mauritianum* were listed in the French pharmacopoeia and constitute one of the best-selling medicinal plants on Reunion Island. Aqueous extracts of *P. mauritianum* have also demonstrated antiviral activity against strains of Zika and Dengue viruses in vitro, without exhibiting genotoxic effects, in several mammalian cell types [9]. The crude acetone extract of *P. mauritianum* was found to harbor antioxidant activity and showed antimicrobial activity, with an MIC of 51 μg/mL recorded against *S. aureus*. Through bioassay guided fractionation, this anti-staphylococcal activity was linked to the presence of corosolic and asiatic acids [10]. Despite its promising biological activities and a large consumption of the leaf infusions by Reunionese people, very few phytochemical studies were found in the literature, and to date, only the two pentacyclic triterpenes mentioned above have been isolated from *P. mauritianum* [8,10].

In an effort to identify new natural antimicrobial compounds and to explore the chemical diversity of plants from Reunion Island, we found that the ethyl acetate extract (EtOAc) from *P. mauritianum* demonstrated strong antimicrobial activity against *S. aureus* (MIC of 8 μg/mL).

## 2. Results and Discussion

Bioassay guided fractionation of the antibacterial EtOAc extract of *Psiloxylon mauritianum* led to isolation of the known molecules Aspidin BB (**1**) [11,12], ursolic acid (**3**) and oleanic acid (**4**) [13], along with compound **2** that had not previously been isolated or described in the literature (Figure 1). The known compounds were identified by comparison of ^1^H and ^13^C data with values reported in the literature, together with crystallography data for **1**. The common triterpenic acids **3** and **4** were isolated as a 6:4 ratio mixture, and the complete ^1^H and ^13^C-NMR assignments were deduced from NMR 1D and 2D experiments conducted on a 700 MHz NMR spectrometer.

Compound **2** was initially obtained as a white amorphous solid. HRESIMS analysis of **2** revealed a molecular formula of C_27_H_36_O_8_ (*m/z* 489.2490 for [M + H]^+^), implying 2 C and 4 H more than in Aspidin BB **1**. The ^1^H-NMR spectrum displayed remarkably downfield-shifted singulet signals at *δ*_H_ 15.86, 11.41 and, 10.05, which are characteristic of the hydroxyl groups found in acylphloroglucinols Aspidin derivatives [14]. The ^1^H-NMR data of **2** were very similar to those for **1** except for the presence of a supplementary signal at *δ*_H_ 1.39 (m) integrating four protons (H11 and H12), and a two methyl triplet at *δ*_H_ 0.98 (*J* = 7.4 Hz) and *δ*_H_ 0.93 (*J* = 7.1 Hz), which are the common signals for methyl terminal groups (Table 1).

These findings suggest that **2** is an analogue of **1** with different structure of the side chains. Interpretation of COSY and HMBC experiments, especially HMBC correlations observed with the two ketonic carbons at *δ*_C_ 208.0 (C8′) and *δ*_C_ 207.5 (C8), easily revealed the presence of valeryl and butyryl chains. The connection of the valeryl side chain to the acylfilicinic acid moiety was determined with HMBC correlation between the protons of the methylene H9 at *δ*_H_ 3.19 with the quaternary carbon C3 at *δ*_C_ 108.9 ppm. The allocation of the butyryl chain was established with the ROESY experiment (Figure 2). In fact, the protons of the methylene H9′ at *δ*_H_ 3.15 displayed a ROE correlation with the protons of the methoxyl H7′ at *δ*_H_ 3.80, which exhibited a clear ROE correlation with the toluene methyl H12′ at *δ*_H_ 2.10.

The structure determined with the NMR data was confirmed by single-crystal X-ray diffraction analysis (See Appendix A). Crystallographic data were recorded at 150 K to reduce agitation induced by the length of the valeryl side chain. Compound **2** was named Aspidin VB, and although its presence has already been mentioned in the literature [14,15], this is the first time that it has been isolated and characterized.

Next, the antimicrobial activity of the crude extract and isolated compounds **1** and **2** was evaluated. First, response against human *Staphylococcus aureus* and MRSA were examined, followed by *Candida albicans* and *Trichophyton rubrum* (Table 2).

Our plant extract exhibited antibacterial activity as well as anticandidal activity with an MIC of 8 µg/mL for *S. aureus* and *C. albicans*.

Compounds **1** and **2** exhibited antibacterial activity against *S. aureus* and MRSA higher than the positive control, with MICs of 0.25 μg/mL for **2** and 2 and 1 μg/mL for **1**, respectively, against these two pathogens. Our results indicated that compound **2** has the same MIC as oxacillin against *S. aureus* and was 16-fold more potent than the standard antibiotic vancomycin against MRSA. Interestingly, Aspidin VB (**2**) showed higher activity than compound **1** against both bacteria. Furthermore, no cytotoxicity was observed for compound **2** at concentrations up to 10^−5^ M (Table 2). In our assays, Aspidin VB (**2**) was more active against MRSA and slightly less toxic than **1**, which is known to exhibit no toxicity when *S. aureus* was killed [16].

In accordance with the literature [16], Aspidin BB (**1**) was strongly active against both *S. aureus* and MRSA but was inactive against the human pathogenic fungi. Aspidin BB (**1**) is known to exert strong antibacterial activity against Gram-positive bacteria, like *S. aureus*, *S. epidermis* or *Propionibacterium acnes* [16,17]. Li et al. identified the relationship between antibacterial activity and increase levels of reactive oxygen species in *S. aureus* cells. Moreover, the authors demonstrated that **1** induced peroxidation of membranes, DNA damage and protein degradation in *S. aureus* [16]. By comparing the effects of compounds **1** and **2** on *S. aureus* strains, our results demonstrated that a longer carbon chain (2 additional carbons) on the acylfilinic acid moiety is correlated with 4- to 8-fold stronger activity (Table 2). Compared to Aspidin BB (**1**), the antibacterial potency of Aspidin VB (**2**) may result from better cell wall penetration due to a longer alkyl chain inducing improved lipophilicity. Computed molecular properties of compounds **1** and **2** were obtained with the SwissADME web tool (http://www.swissadme.ch), and the results are detailed in Appendix A.

## 3. Materials and Methods

### 3.1. General Experimental Procedures

Nuclear magnetic resonance (NMR) spectra were recorded on a Bruker 500 MHz spectrometer or on a Bruker 700 MHz spectrometer equipped with 5 mm inverse detection Bruker. Chemical shifts (δ) are reported in ppm based on the signal for TMS. Chemical shifts were referenced using the corresponding solvent signals (*δ*_H_ 2.05 and *δ*_C_ 29.92 for (CD_3_)_2_CO). HRESIMS measurements were performed using a Waters Acquity UHPLC system with a column bypass coupled to a Waters Micromass LCT Premier time-of-flight mass spectrometer equipped with an electrospray interface (ESI). X-ray diffraction data for compound **1** were collected on the PROXIMA 2A (PX2A) beamline at the SOLEIL Synchrotron, Gif-sur-Yvette, Paris, France. They were indexed, integrated with XDS [18] and scaled with AIMLESS [19], as implemented within the autoProc toolbox [20]. For compound **2**, data were collected using redundant ω scans on a Rigaku XtaLabPro single-crystal diffractometer using microfocus Mo Kα radiation and a HPAD PILATUS3 R 200K detector. Its structure was readily solved by intrinsic phasing methods (SHELXT) [21] and by full-matrix least-squares methods on F2 using SHELX-L [22]. The non-hydrogen atoms were refined anisotropically, and hydrogen atoms, identified in difference maps, were positioned geometrically and treated as riding on their parent atoms. Molecular graphics were computed with Mercury 4.3.0. Flash chromatography was performed on a Grace Reveleris system with dual UV and ELSD detection equipped with a 40 g C_18_ column. Preparative HPLCs were conducted with a Gilson system equipped with a 322 pumping device, a GX-271 fraction collector, a 171 diode array detector, and a prepELSII detector electrospray nebulizer. The columns used for these experiments included a Phenomenex Kinetex C8 5 μm 4.6 × 250 mm analytical column and Phenomenex Kinetex C8 5 μm 21.2 × 250 mm preparative column. The flow rate was set to 1 or 21 mL/min, respectively, using a linear gradient of H_2_O mixed with an increasing proportion of CH_3_CN. Both solvents were of HPLC grade, modified with 0.1% formic acid.

### 3.2. Plant Material

Leaves of *Psiloxylon mauritianum* Baill. (Myrtaceae) were collected in Les Avirons, Reunion Island in 2016, identified by Raymond Lucas (Association APN, Réunion). A voucher specimen was deposited at ICSN-CNRS.

### 3.3. Extraction and Isolation

After collection, plant material was air dried in the shade at room temperature. Crushed dried leaves (135 g) were extracted by maceration with EtOAc (2 × 0.7 L, 2 × 24 h) on a rotary shaker (90 rpm). The organic solvent was collected by vacuum filtration and concentrated to dryness under reduced pressure to yield 10.6 g of extract. A portion of the extract (1.2 g) was subjected to reverse phase flash chromatography using a gradient of H_2_O mixed with an increasing proportion of CH_3_CN, both with 0.1% formic acid, to afford 14 fractions (A to N). A portion of fraction I (10 mg), eluted with 100% CH_3_CN, was subjected to preparative HPLC (isocratic elution at 20:80) to afford the mixture ursolic acid **3**:oleanic acid **4** (6:4) (2.4 mg, RT = 6.0 min), and Aspidin BB **1** (4.3 mg, RT = 12.5 min). A portion of fraction J (40 mg) was washed with cold MeOH (0 °C) to afford Aspidin VB **2** (20 mg). After NMR experiments, small crystal needles were observed in samples tubes of Aspidin BB **1** and Aspidin VB **2**. These crystals were carefully collected and analyzed by X ray crystallography.

*Aspidin BB* (**1**): White amorphous solid or colorless crystal needles; ^1^H-NMR (500 MHz, (CD_3_)_2_CO: *δ*_H_ 3.80 (3H, s, H7′), 3.57 (2H, s, H7), 3.18 (2H, dd, *J* = 7.3, 7.3 Hz, H9), 3.15 (2H, dd, *J* = 7.2, 7.2 Hz, 2H), 2.10 (3H, s, H12′), 1.70 (4H, m, H10, H10′), 1.49 (6H, s, H12, H13), 1.00 (3H, t, *J* = 7.4 Hz, H11), 0.98 (3H, t, *J* = 7.4 Hz, H11′); ^13^C-NMR (125 MHz, (CD_3_)_2_CO: *δ*_C_ 208.0 (C8′), 207.4 (C8), 199.8 (C4), 188.4 (C2), 172.7 (C6), 163.6 (C6′), 161.5 (C4′), 160.3 (C2′), 113.1 (C5′), 111.8 (C1), 110.2 (C1′), 109.0 (C3), 108.5 (C3′), 62.1 (C7′), 45.0 (C5), 44.4 (C9′), 43.5 (C9), 25.1 (C12), 25.1 (C13), 18.8 (C10), 18.2 (C10′), 17.7 (C7), 14.2 (C11), 14.1 (C11′), C12′ (9.5); HRESIMS [M + H]^+^
*m/z* 461.2173 (calc. for C_25_H_33_O_8_, 461.2175).

*Aspidin VB* (**2**): White amorphous solid or colorless crystal needles; ^1^H-NMR (500 MHz, (CD_3_)_2_CO and ^13^C-NMR (125 MHz, (CD_3_)_2_CO) see Table 1; HRESIMS [M + H]^+^
*m/z* 489.2490 (calculated for C_27_H_37_O_8_, 489.2488).

*Ursolic acid* (**3**): *oleanic acid* (**4**) *(6:4 mixture)*: White amorphous; ^1^H-NMR (700 MHz, (CD_3_)_2_CO and ^13^C-NMR (175 MHz, (CD_3_)_2_CO) see Table 1; HRESIMS [M + H]^+^
*m/z* 457.3661 (calculated for C_30_H_49_O_3_, 457.3682).

### 3.4. Determination of Minimal Inhibitory Concentration

The crude extract and pure compounds isolated were tested against human pathogenic microorganisms, including the bacterium *Staphylococcus aureus* (ATCC 29213), MRSA (ATCC 33591), *Candida albicans* (ATCC 10213) and *Trichophyton rubrum* (SNB-TR1) to screen their antibacterial and antifungal activities. All ATCC strains were purchased from the Pasteur Institute. The clinical isolate was provided by Prof. Philippe Loiseau, Université Paris Sud. The ITS sequence was deposited in the NCBI GenBank database under the registry number KC692746 corresponding to SNB-TR1 strain. The tests were conducted in accordance to the reference protocols from the European Committee on Antimicrobial Susceptibility Testing [23,24,25,26]. The standard microdilutions, ranging from 256 to 0.25 μg/mL were made from stock solutions prepared in DMSO (Sigma-Aldrich, France). The microplates were incubated at 35 °C, and MIC values were obtained after 48 h for *C. albicans*, 24 h for *S. aureus* and 5 days for *T. rubrum*. The MIC values reported in Table 2 refer to the lowest concentration preventing visible growth in the wells. Vancomycin (Sigma-Aldrich, Saint-Quentin Fallavier, France) and oxacillin (Sigma-Aldrich, Saint-Quentin Fallavier, France) were used as positive controls for bacteria. Fluconazole (Sigma-Aldrich, Saint-Quentin Fallavier, France) and itraconazole (Sigma-Aldrich, Saint-Quentin Fallavier, France) were used as positive controls for fungi. All assays were conducted in duplicate.

### 3.5. Cytotoxicity Evaluation

Human lung fibroblast cells (MRC-5) were purchased from ATCC (Rockville, MD, USA) and cultured as recommended. Cell growth inhibition was determined by an MTS assay according to the manufacturer’s instructions (Promega, Madison, WI, USA). The cells were seeded in 96-well plates containing the growth medium. After 24 h of culture, samples were dissolved in DMSO (Sigma-Aldrich, France), and added to the cells (at 1 and 10 μM final concentrations). After 72 h of incubation, the reagent was added, and the absorbance at 490 nm was recorded using a plate reader. Cell viability was evaluated in comparison with untreated control cultures. Docetaxel (Taxotère) was used as positive control (IC_50_: 0.5 nM). All assays were conducted in triplicate.

## 4. Conclusions

Chemical investigation of the ethyl acetate extraction of *P. mauritianum* leaves led to the first report of the known compounds Aspidin BB (**1**), ursolic acid (**3**) and oleanic acid (**4**) in this plant. We also reported the presence of Aspidin VB (**2**), an acylphloroglucinol derivative never before described. The structure of compound **2** was determined by X-ray and NMR analyses. Aspidin VB (**2**) exhibited very strong activity against bacteria *S. aureus* and MRSA and showed no toxicity at 10^−5^ M in an MRC5 cell line. The two acylphloroglucinols compounds, **1** and **2**, seem to be responsible for the antibacterial activity identified in the crude extract. Due to the biological activities of Aspidin VB, the antibacterial potency of **2** will be evaluated against different resistant *S. aureus* strains and additional pathogenic bacteria. This study further demonstrates the importance of phytochemical studies on medicinal plants from Reunion Island, which are largely unexplored.

## Figures and Tables

**Figure 1 molecules-25-03565-f001:**
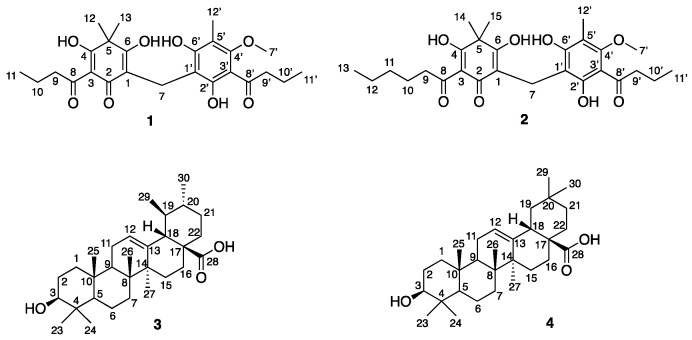
Structure of the compounds isolated from *Psiloxylon mauritianum*.

**Figure 2 molecules-25-03565-f002:**
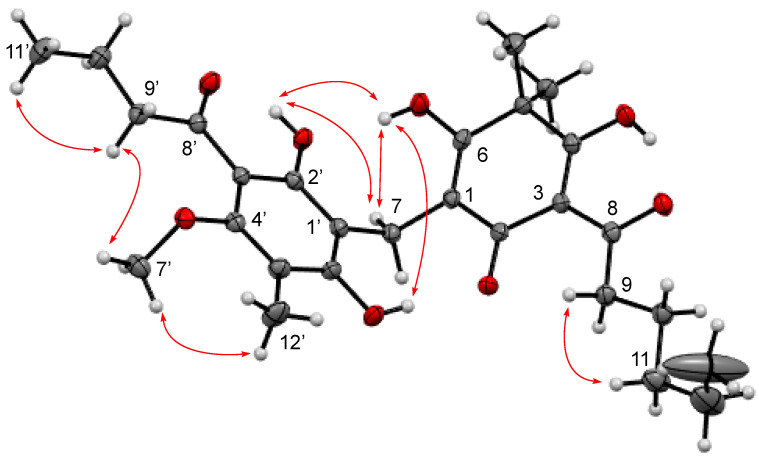
Key ROESY correlations in Aspidin VB (**2**).

**Table 1 molecules-25-03565-t001:** The 1D and 2D NMR data for Aspidin VB (**2**) in acetone-*d*_6_.

Position	Aspidin VB				
	*δ* _C_ ^1^	*δ*_H_*(J* in Hz) ^2^	COSY	HMBC	ROESY
1	111.9				
2	188.4				
3	108.9				
4	199.9				
5	45.1				
6	172.7				
7	17.7	3.57, s		C1, C2, C6, C1′, C2′, C6′	
8	207.5				
9	41.6	3.19, dd (7.2, 7.2)	H8	C3, C8, C10, C11	H11
10	25.3	1.67, m	H9, H11	C8, C9, C11, C12	
11	32.5	1.39, m	H10, H12	C12	H9
12	23.2	1.38, m	H11, H13	C11	
13	14.3	0.93, t (7.1)	H12	C11, C12	
1′	110.2				
2′	160.6				
3′	108.5				
4′	161.5				
5′	113.2				
6′	163.6				
7′	62.1	3.80, s		C4′	H9′
8′	208.0				
9′	44.8	3.15, dd (7.2, 7.2)	H10′	C8′, C10′, C11′	H7′
10′	18.8	1.72, sex (7.4)	H9′, H11′	C8′, C10′, C11′	
11′	14.2	0.98, t (7.4)	H10′	C9′, C10′	H9′
12′	9.5	2.10, s		C4′, C5′, C6′	H7′
6-OH		10.05, s		C1, C5, C6	H7, 2′-OH, 6′-OH
2′-OH		15.86, s		C1′, C3′, C8′	6-OH
6′-OH		11.41, s		C1′, C5′, C6′	H7, 6-OH

^1^ Recorded at 500 MHz. ^2^ Recorded at 125 MHz.

**Table 2 molecules-25-03565-t002:** Antimicrobial and cytotoxic results for EtOAc crude extract and isolated compounds.

Compounds	MIC (µg/mL)				MRC5 Cell Viability (%)	
	*C. albicans* ATCC 10213	*T. rubrum* SNB-TR1	*S. aureus* ATCC 29213	MRSA ATCC 33591	10^−5^ M	10^−6^ M
**1**	>256	>256	2	1	86 ± 3	104 ± 1
**2**	>256	256	0.25	0.25	99 ± 2	105 ± 2
Crude extract	8	256	8	nd	96 ± 2	100 ± 3
Fluconazole ^1^	1	4	nd	nd	nd	nd
Itraconazole ^1^	<0.5	<0.5	nd	nd	nd	nd
Oxacillin ^1^	nd ^2^	nd	0.25	nd	nd	nd
Vancomycin ^1^	nd	nd	nd	4	nd	nd

^1^ Positive control. ^2^ Not determined.

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
