# Peer review of "Potent and Non-Cytotoxic Antibacterial Compounds Against Methicillin-Resistant Staphylococcus aureus Isolated from Psiloxylon mauritianum, A Medicinal Plant from Reunion Island"

_molecules, 2020, doi:10.3390/molecules25163565_

Round 1
Reviewer 1 Report
The article presented by Sorres et al. explores the antimicrobial activity of both crude extract and isolated molecules from the plant Psiloxylon mauritianum on clinical S. aureus and MARSA bacteria. The widespread use of antibiotics for the treatment of bacterial infections has propagated the emergence and spread of antibiotic resistant pathogens, thus pushing the development and use of alternative, non-antibiotic therapies. The authors found that while crude extract from P. mauritianum was effective at hindering both S. aureus and MARSA growth, two purified components of the extract had greater efficacy than the standard and commonly used antibiotic against the MARSA strain. In addition, there seemed to be very little cytotoxicity to human cell lines; however, the two components were less effective at limiting the growth of common fungal pathogens. The structure of these two components were predicted and validated, and one was found to be a novel structural derivative of the other that has never been reported in the literature. This article is concise, well written and provides information about a novel compound that has potential to supplement or replace common antibiotics used by struggling healthcare systems that are faced with the challenge of combating antibiotic resistant bacteria. Apart from a few minor comments I would recommend this work for publication
Minor comments
Line 25 : replace “mg/ml” to “ug/ml” (at least according to Table 2)
Line 40: add citation to the end of sentence
Line 47: add citation to the end of sentence
Line 63: change “….ethyl acetate extract….” To “….ethyl acetate (EtOAc) extract….”
Line 74: in the Figure 1 figure legend, change “Psyloxilon mauritian” to “Psiloxylon mauritianum” to be consistent with the way it is spelled elsewhere
Author Response
All the modifications listed below have been added in the manuscript
Comments/ :
- Line 25 : replace “mg/ml” to “ug/ml” (at least according to Table 2)
Done
- Line 40: add citation to the end of sentence
Done
- Line 47: add citation to the end of sentence
Done
- Line 63: change “….ethyl acetate extract….” To “….ethyl acetate (EtOAc) extract….”
Done
- Line 74: in the Figure 1 figure legend, change “Psyloxilon mauritian” to “Psiloxylon mauritianum” to be consistent with the way it is spelled elsewhere
Done
Reviewer 2 Report
The paper by Sorres et al. describes a new anti-staphylococcal drug class called aspidins. Anti-bacterial activity of two aspidin compounds showed very good efficacy against two strains of Staphylococcus aureus with very low cytotoxicity. Overall, the paper is organized well. However, several changes in the materials and methods need to be addressed.
- The abstract needs a couple of changes. Line 17 should have antibiotic in front of resistant and Line 25 should be 0.25 ug/mL not 0.25 mg/mL.
- Line 56 should not state strong antimicrobial activity is the MIC is only 51 ug/mL.
- In the materials and methods section, you need to elaborate more on the details of the biological activities. You should split the biological activities into two subheadings: MIC and cytotoxicity. What are your compounds suspended in for testing? What incubation temp did you use for your MICs? What are the sources for the drugs that you used? It is oxacillin, not oxaciline. How many replicates were done? Too few details were provided for doing the cytotoxicity. Did you do both the neutral red and MTT assays? Controls? How many replicates were done?
- Run MICs for oxacillin and vancomycin against both strains of S. aureus.
- What is the solubility of both compounds?
- You have only run your analysis against two strains of S. aureus, which is quite limited.
- All of the references should be one style.
Author Response
Thank you for your comments, the informations requested have been completed in the manuscript and is detailed below.
Comments/ answers :
- The abstract needs a couple of changes. Line 17 should have antibiotic in front of resistant
Done
and Line 25 should be 0.25 ug/mL not 0.25 mg/mL.
Done
- Line 56 should not state strong antimicrobial activity is the MIC is only 51 ug/mL.
We agree with your comment and so we have removed the term ‘strong’. Originally, the term is issued from the abstract of Rangasamy et al., who have written: “PM was found to be both strongly antibacterial and antioxidant”.
- In the materials and methods section, you need to elaborate more on the details of the biological activities. You should split the biological activities into two subheadings: MIC and cytotoxicity. What are your compounds suspended in for testing? What incubation temp did you use for your MICs? What are the sources for the drugs that you used? It is oxacillin, not oxaciline. How many replicates were done? Too few details were provided for doing the cytotoxicity. Did you do both the neutral red and MTT assays? Controls? How many replicates were done?
As recommended, the biological activities method was split and augmented. The used methods are briefly exposed, but they are fully detailed in cited references (EUCAST 2003 and EUCAST 2008). The sections answer now to all of your relevant issues.
- Run MICs for oxacillin and vancomycin against both strains of S. aureus.
In our test both oxacillin and vancomycin are used as positive controls. In general, oxacillin is recommended for sensible strain ATCC 29213 and vancomycin for MRSA. In fact, in the Table 3 of the EUCAST paper (2003) are depicted the MICs of oxacillin (0.25 ug/ml) and vancomycin (1 ug/ml) against ATCC 29213 strain. Clearly, oxacillin is a better positive control than vancomycin on this strain. In addition, Methicillin-resistant Staphylococcus aureus strains are generally resistant to oxacillin (It is quite logic because both oxacillin and methicillin belong to β-lactams antibiotic class). Basri and Sandra (http://dx.doi.org/10.1155/2016/5249534) found a MIC of 31.25 ug/ml against MRSA ATCC 33591 (table 1 of the paper). So, oxacillin was not suitable for the positive control in the MIC evaluations against ATCC 33591 strain. A table footnote was added for clarification.
- What is the solubility of both compounds?
Compounds 1 and 2 are well soluble in acetone and in methanol at room temperature. Due to available quantities, the solubility in water was not determined. Calculated physicochemical descriptors from SwissADME web service (http://www.swissadme.ch/index.php) were added in Supplementary materials Figures S14 and S15 and cited in the text line 132.
- You have only run your analysis against two strains of S. aureus, which is quite limited.
It is sure that antimicrobial potential of aspidin VB must be fully evaluated. Collaborations are under constructions for testing MICs against clinical strains and we hope that results will be soon published in Molecules.
- All of the references should be one style.
References 25 and 26 were adapted.
Round 2
Reviewer 2 Report
The authors have addressed all of my comments.